# Study by Optical Spectroscopy of Bismuth Emission in a Nanosecond-Pulsed Discharge Created in Liquid Nitrogen

**DOI:** 10.3390/molecules26237403

**Published:** 2021-12-06

**Authors:** Anna V. Nominé, Cédric Noel, Thomas Gries, Alexandre Nominé, Valentin A. Milichko, Thierry Belmonte

**Affiliations:** 1Laboratoire d’Interaction Plasma Extrême Surface du CNRS, l’Institut Jean Lamour, Université de Lorraine, F-54000 Nancy, France; anna.nomine@univ-lorraine.fr (A.V.N.); c.noel@univ-lorraine.fr (C.N.); thomas.gries@univ-lorraine.fr (T.G.); alexandre.nomine@univ-lorraine.fr (A.N.); 2Department of Nanophotonics and Metamatarials, ITMO University, 49 Kronverkskii pr., Saint. Petersburg 197101, Russia; ariesval@mail.ru

**Keywords:** optical emission spectroscopy, discharges in liquids, bismuth Stark parameters

## Abstract

Time-resolved optical emission spectroscopy of nanosecond-pulsed discharges ignited in liquid nitrogen between two bismuth electrodes is used to determine the main discharge parameters (electron temperature, electron density and optical thickness). Nineteen lines belonging to the Bi I system and seven to the Bi II system could be recorded by directly plunging the optical fibre into the liquid in close vicinity to the discharge. The lack of data for the Stark parameters to evaluate the broadening of the Bi I lines was solved by taking advantage of the time-resolved information supported by each line to determine them. The electron density was found to decrease exponentially from 6.5 ± 1.5 × 10^16^ cm^−3^ 200 ns after ignition to 1.0 ± 0.5 × 10^16^ cm^−3^ after 1050 ns. The electron temperature was found to be 0.35 eV, close to the value given by Saha’s equation.

## 1. Introduction

Processes based on discharges in liquids are commonly used to synthesize original nanoobjects with high yields [1,2,3,4,5]. In liquid nitrogen, they are particularly interesting, as they lead to the synthesis of two-dimensional structural materials like nanosheets [6,7,8], exhibiting high levels of selectively exposed reactive facets and the fast separation of electrons and holes.

Bismuth trioxide is a semiconducting material with high ionic conductivity and photosensitivity, a large energy bandgap and a high refractive index, making them useful for various applications such as optical coatings, solar or fuel cells, microwave-integrated circuits and/or gas sensors [9,10,11,12]. Its synthesis as nanosheets by discharges in liquid nitrogen and further oxidation to air has been demonstrated recently.

In the present work, attention is paid to optical emissions during the discharge process. By applying a nanosecond-pulsed high voltage to bismuth electrodes submerged in liquid nitrogen, the erosion process leads to the production of metallic nanosheets. After processing, liquid nitrogen evaporates, and the nanosheets, exposed to air, get oxidized and are transformed into Bi_2_O_3_. Diagnosing these discharges is not an easy task, as they are very small (tens of micrometres) and pretty fast (hundreds of nanoseconds) [13]. However, such investigations are essential to assess the plasma parameters, which is important to improve our control of the process [14,15,16].

Basic data concerning line broadening for bismuth has been relatively unexplored. For instance, if Stark parameters are available for Bi II transitions [17,18,19], they are missing for Bi I transitions. Discharges in liquids are close to the local thermodynamic equilibrium. They are characterised by high-electron densities (typically around 10^16^–10^17^ cm^−3^), low electron temperatures (below 1 eV) and they are submitted to high pressures (up to several tens of bars in liquid nitrogen). Thus, they are optically thick at inception and lead to non-symmetrical line shapes, self-absorbed transitions and line shifts in the wavelength [13].

Methods based on the so-called “six free parameters deconvolution” (SFPD) procedure were developed to determine from the experimental data the parameters of the given emission lines [20,21,22,23,24]. Here, the broadening parameters of the selected bismuth lines are determined together with the plasma parameters. We also propose a new way to improve the quality of the raw signals recorded during measurements by introducing the optical fibre directly in the liquid, as close to the discharge as possible.

## 2. Materials and Methods

The experimental set-up was presented in detail in Reference [25]. Briefly, a pin-to-pin electrode configuration was used. Electrodes were bismuth rods (Goodfellow France, Lille, France, 5 mm in diameter—99.9% purity) cut to sharp points (curvature radius: 100 µm—see Figure 1). A high DC voltage power supply (Technix, Créteil, France, SR15-R-1200–15 kV–80 mA) fed a solid-state switch (Behlke, Kronberg im Taunus, Germany, HTS-301-03-GSM) connected to a one-pin electrode, the other electrode being grounded. The voltage rise time was 20 ns without a ballast resistor. The applied voltage was +10 kV, and the current reached a maximum of 60 amperes. The voltage pulse was is 75 ns. The energy dissipated in the average per pulse was between 1 and 5 mJ, typically.

Optical emission spectroscopy was performed with a 550-mm focal length monochromator (Jobin–Yvon TRIAX 550) equipped with a 100-gr. mm^−1^ grating blazed at 500 nm for the overall spectra in the visible range (250–900 nm). For the time evolution of specific lines, an 1800-gr. mm^−1^ grating blazed at 330 nm was used also in the range 250–900 nm. It was coupled with a HORIBA Jobin–Yvon i-Spectrum Two iCCD detector. The iCCD detector was triggered by the trigger output of the oscilloscope, which was itself triggered by the current peak. Each measurement was averaged over 50 spectra recorded with an exposure time of 50 ns. The iCCD gain was always set to its maximum. Although discharges in dielectric liquids are known to be stochastic, using a solid-state switch with a 20-ns rise time ensures a high level of reproducibility, because breakdown necessarily occurs within a time window inferior to the exposure time. However, because of the generation of high-frequency signals by discharge current oscillations (at about 1 MHz), ghost lines are sometimes observed. They are easily identified in time-resolved data, since they disappear from one spectrum to the other.

In this work, light was collected through a multi-strand optical fibre. Due to the weakness of the emitted light when the fibre was out of the liquid, a decision was made to strip the metallic tip of the fibre and to plunge the nonconductive strands, tied together by Teflon tape, directly into liquid nitrogen as close to the discharge as possible. Processing this way enables a strong increase in the line intensity and makes ionic lines (Bi II system) appear in the spectra (Figure 2). All ionic lines but the one at 430.17 nm were nonetheless very weak and only visible from 200 to 350–400 ns after breakdown with the 100-gr. mm^−1^ grating. Though nanoparticles are produced by electrode erosion, deposition on the fibre was negligible. It is certainly because of shockwaves emitted during the very early stage of the discharge process that push nanoparticles away from the interelectrode gap.

Transitions belonging to the Bi I and Bi II systems were identified during this study. They are listed in Table 1 (see Appendix A for the visualization of transitions in the corresponding Grotrian diagrams) [26,27]. The last three columns contained data evaluated for the present study from the experimental spectra. The two previous columns were theoretical estimates of *C’*_3_ and *C’*_6_ coefficients. No line beyond 500 nm was observed. Some lines were too weak or too noisy to be accurately treated. Lines were considered as being weak with regards to their intensity as recorded by the selected optical set-up. Thus, the UV transition at 262.79 nm was strong with the 100-gr. mm^−1^ grating but weak with the more dispersive 1800-gr. mm^−1^ grating. As a high resolution was needed for an accurate description, only strong or very strong (see Table 1) transitions recorded with the 1800-gr. mm^−1^ grating were modelled, weak or very weak transitions being however considered when mixed with strong lines.

It was not possible in the present conditions to exploit the line shifts. Theoretically, at the highest electron density found in this work (~6–7 × 10^16^ cm^−3^), i.e., when the lines were strongly broadened and exhibited flat or reversed maxima, the expected line shift was, at best, 80 pm. With the 1800-gr. mm^−1^ grating used in this work, the distance between two consecutive pixels of the detector corresponded to a wavelength difference of 23 pm.

The lack of nitrogen transition lines in the recorded spectra was attributed the optical thickness of the medium, as discussed in Reference [2]. The ground states were highly populated, leading to light trapping in the medium. The lower states of the bismuth transitions, conversely, were weakly populated states, enabling the exit of light out of the medium.

## 3. Theory

The theoretical aspects supporting the present study have been described in detail in two publications of ours [13,28]. A short version, presenting only the elements required for the present work, is provided for the sake of clarity.

Basically, optical transitions in spark discharges in liquids are broadened by various phenomena whose relative weights change in time because of the rapid evolution of the most important parameters that control the discharge. The electron temperature, the electron density, the total pressure and the medium optical thickness are known to strongly vary at the very beginning of the process, i.e., from 0 to ~20–25 ns, more moderately from 25 to 150–200 ns when the current drops and then only weakly beyond 200 ns when it oscillates.

Before describing the mechanisms responsible for line broadening, an optical model is needed to account for the trapping of a part of the emitted light.

### 3.1. Optical Model

When a photon is emitted by an electronic transition between an upper level and a lower level, if it travels through a medium in which the concentration of the species standing in the lower level is high enough, it will be absorbed and emitted again several times. The trapping probability depends on the wavelength of the photon. If it matches the energy transition, i.e., if it has a wavelength corresponding to the line centre rather than the line wings, it will be higher. This leads to a depletion in the photons and a hole in the centre of the energy distribution (see Appendix A).

The spectral density of the radiant flux (expressed in W m^−2^ Hz^−1^ sr^−1^) depends on the photon frequency *ν* and on the distance from the discharge centre *r*:(1)dI(r,ν)=(ε(r,ν)−κ(r,ν)I(r,ν))dr

The absorption κ(r,ν) (m^−1^) and emission ε(r,ν) (W m^−3^ Hz^−1^ sr^−1^) coefficients are determined from Einstein’s coefficients for spontaneous emission *A_ul_*, photo-absorption *B_lu_* and induced emission *B_ul_*:(2)$$\varepsilon \left(r,\nu \right)=\frac{A_{ul}n_{u}\left(r\right)h\nu_{0}}{4\pi }f\left(r,\nu \right)$$
(3)$$\kappa \left(r,\nu \right)=\frac{\left[B_{lu}n_{l}\left(r\right)-B_{ul}n_{u}\left(r\right)\right]h\nu_{0}}{c}f\left(r,\nu \right),$$
where $n_{l}\left(r\right)$ and $n_{u}\left(r\right)$ are the densities of the lower and upper levels of the transition, and f(r,ν) is the normalized spectral distribution for an atomic transition at a given position *r*: $\int_{-\infty }^{+\infty }f\left(r,\nu \right)d\nu =1$.

In discharge in liquids, Planck’s emission dominates at the beginning of the process. This specific contribution, denoted as $\varepsilon_{bg}\left(r,\nu \right)$, superimposes upon the line emission, and Equation (2) changes to:(4)$$\varepsilon \left(r,\nu \right)=\frac{A_{ul}n_{u}\left(r\right)h\nu_{0}}{4\pi }f\left(r,\nu \right)+\varepsilon_{bg}\left(r,\nu \right).$$

The transition probability for the induced emission is given by $B_{ul}={\left(c/\nu_{0}\right)}^{3}\left(A_{ul}/8\pi h\right),$ and it is related to *B_lu_* by $B_{lu}=\left(g_{u}/g_{l}\right)B_{ul},$ where *g_u_* and *g_l_* are the degeneracy degrees of the upper and lower levels. The intensity of the radiant flux is obtained by the integration of Equation (1) over the photon path:(5)$$I\left(\nu \right)=\frac{1}{exp\left[-\int_{-\infty }^{+\infty }\kappa \left(r,\nu \right)dr\right]}\int_{-\infty }^{+\infty }\exp\left[-\int_{-\infty }^{+\infty }\kappa \left(r,\nu \right)dr\right]\varepsilon \left(r,\nu \right)dr.$$

Therefore, a spatial density distribution is needed. For electrons and neutral species, a Gaussian distribution is commonly chosen for discharges exhibiting a cylindrical symmetry [29,30]:(6)$$n_{e}\left(r\right)=N_{e}\left(0\right)\exp\left(\frac{-r^{2}}{\sigma_{e}^{2}}\right);\;\;n_{u}\left(r\right)=N_{u}\left(0\right)\exp\left(\frac{-r^{2}}{\sigma_{u}^{2}}\right);\;\;n_{l}\left(r\right)=N_{l}\left(0\right)\exp\left(\frac{-r^{2}}{\sigma_{l}^{2}}\right).$$

The standard deviations may be specific to each species. For the sake of simplicity, and to decrease as much a possible the number of parameters in the model, we set: $\sigma_{e}^{2}=\sigma_{u}^{2}=\sigma_{l}^{2}$. Assuming a Boltzmann distribution between the energy levels leads to: $\frac{N_{u}\left(0\right)}{N_{l}\left(0\right)}=\frac{g_{u}}{g_{l}}\exp\left(\frac{-h\nu_{0}}{k_{B}T_{e}}\right)$ (Appendix A). Such profiles are not physical, because they do not take into account the radial gradients of the electron temperature and gas pressure. As these data are unknown in the present condition, the simplest description was adopted to account for the optical thickness of the emission lines.

### 3.2. Line Broadening

Due to the heavy mass of bismuth (M = 208.98 u), line broadening mechanisms are essentially limited to resonance broadening for the only transition at 306.77 nm and to instrumental and Stark broadenings for all transitions. Indeed, Doppler and van der Waals (both in the quasistatic and impact approximations) broadenings remain at negligible levels, these contributions being systematically evaluated (see Reference [13] for the formula used in that purpose). In Table 2, the various possible contributions to broadening for different lines are given for *n_e_* = 6 × 10^16^ cm^−3^. Three main contributions are to be considered: instrumental, resonance and Stark broadenings.

The instrumental broadening was found equal to 0.10 nm thanks to a calibration using a He-Ne laser. The line shape thus obtained was approximately Gaussian.

The transition at 306.77 nm was resonant, as it involved an upper level directly dipole-coupled to the ground state (the electric dipole transition and the corresponding force derives from the potential $\Delta V_{3}=\hbar C_{3,gl}/r^{3}$). At low densities, lines broadening by resonance adopt a Lorentzian shape. The full width at half-maximum (FWHM) is given by:(7)$$\Delta \nu_{res}=\pi k_{gl}C_{3,gl}^{\prime }\sqrt{\frac{g_{g}}{g_{l}}}N_{g}.$$

*k_gl_* is a numerical constant for the transition between the lower level and the ground level. We used *k_gl_* = 1.53 [31]. *N_g_* is the density of the ground state. The coefficient $C_{3,gl}^{\prime }\;\;\left(m^{3}\cdot s^{-1}\right)$ can be evaluated from the oscillator strength *f_gl_* with:(8)$$C_{3,gl}^{\prime }=\frac{C_{3,gl}}{\hbar }=\frac{1}{4\pi }\frac{1}{4\pi \varepsilon_{0}}\frac{e^{2}}{m_{e}}\frac{f_{gl}}{\nu_{lg}},$$
and: (9)$$f_{gl}=\frac{C_{3,gl}}{\hbar }=\frac{4\pi \varepsilon_{0}}{8\pi^{2}}\frac{c^{3}m_{e}}{e^{2}\nu_{lg}^{2}}\frac{g_{l}}{g_{g}}A_{lg}.$$

The constants have their usual meanings.

The $C_{6}^{\prime }$ coefficients were also estimated (see Table 1 and Reference [28]) using the following expression:(10)$$C_{6}^{\prime }=\frac{2\pi e^{2}}{4\pi \varepsilon_{0}h}\alpha \left|\langle R_{u}^{2}\rangle -\langle R_{l}^{2}\rangle \right|a_{0}^{2}$$
where *α* is the polarizability of the emitter, $\langle R_{u}^{2}\rangle$ the mean square radius of the atom in its excited level *j* and *a*_0_ the Bohr radius.

Stark profiles are produced under the action of high-frequency fields of electrons (the linear Stark effect obeys a potential $\Delta V_{2}=-\hbar C_{2}/r^{2}$) and under the action of the low-frequency fields of ions (the quadratic Stark effect with $\Delta V_{4}=-\hbar C_{4}/r^{4}$). In the case of non-hydrogenoid atomic transition, ion broadening is not negligible, and line profiles become asymmetric, as was clearly the case with bismuth. Within the quasistatic approximation, the line profile is given by:(11)$$j_{A,R}\left(x\right)=\frac{1}{\pi }\int_{0}^{\infty }\frac{H_{R}\left(\beta \right)}{1+{\left(x-A^{4/3}\beta^{2}\right)}^{2}}d\beta ,$$
where $x=\frac{\lambda -\lambda_{0}-d_{e}}{w_{e}}$, *w_e_* is the half-width at the half-maximum (HWHM) of the Stark broadening due to electron collision and *d_e_* is the corresponding shift. *H_R_*(*β*) is the distribution function of the micro-field intensity that depends on the normalised intensity of the Holtsmark field *β* = *F*/*F*_0_, *F*_0_ being the intensity of the normal field. *A* is the broadening parameter due to the static ions. *R* is the ratio of the average distance between the ions and the Debye radius, i.e., the screening parameter of Debye:(12)$$R={\left[\frac{36\pi e^{6}n_{e}}{{\left(k_{B}T_{e}\right)}^{3}}\right]}^{1/6}\approx 0.0899\frac{{\left(n_{e}\left[cm^{-3}\right]\right)}^{1/6}}{T_{e}^{1/2}}.$$

The expression of $j_{A,R}\left(x\right)$ is determined thanks to the Woltz tables [32] from *w_e_*. When the ion dynamics are no longer negligible, it is necessary to introduce a correction factor. The criterion to fulfil is then [20]:(13)$$B=A^{1/3}\frac{8.06\times 10^{-7}w_{e}^{ref}\left[nm\right]}{{\left(\lambda \left[nm\right]\right)}^{2}}{\left(n_{e}\left[m^{-3}\right]\right)}^{2/3}\sqrt{\frac{\mu \left[amu\right]}{T_{g}\left[K\right]}}<1,$$
where $w_{e}^{ref}$ is the value of *w_e_* when *n_e_* = 10^23^ m^−3^. *µ* is the reduced mass of the ion or neutral perturber in arbitrary mass units. This means that strong collisions due to electrons and ions are well-separated in time. Then, the total broadening is given by:(14)$$\Delta \lambda_{s}=2w_{e}\left[1+1.75AD_{J}\left(1-0.75\kappa R\right)\right]$$
with $D_{J}=\frac{1.36}{1.75\left(1-0.75R\right)}B^{-1/3}$ if $B<{\left(\frac{1.36}{1.75\left(1-0.75R\right)}\right)}^{3}$ or $D_{J}=1$ if $B\geq {\left(\frac{1.36}{1.75\left(1-0.75R\right)}\right)}^{3}$ [20]. In the latter case, the ion dynamics are negligible, and the line shape is treated by considering the quasistatic approximation.

Approximate formulas, resorting to parametric functions, were derived for easier use. They were reasonably accurate if 0.05≤α≤0.5 and r≤0.8, these two quantities being defined hereafter. The line width broadened by the Stark effect is then given by:(15)$$\Delta \lambda_{s}=2\left[1+1.75\alpha \left(1-0.75\kappa r_{D}\right)\right]w,$$
where *w*(*T_e_*, *n_e_*) and *α*(*T_e_*, *n_e_*) are two parametric functions tabulated by Reference [33]. They represent the electronic contribution expressed in wavelengths and the ion quasi-static-broadening parameter. They scale, respectively, as $n_{e}$ and $n_{e}^{1/4}$. Therefore, they are more conveniently written as $w\left(T_{e},n_{e}\right)=\frac{n_{e}}{n_{e}^{0}}w_{e}\left(T_{e}\right)$ and $\alpha \left(T_{e},n_{e}\right)={\left(\frac{n_{e}}{n_{e}^{0}}\right)}^{1/4}a\left(T_{e}\right),$ where *w_e_* and *a* are dependent on *T_e_* only. $n_{e}^{0\;}$= 10^22^ m^−3^. $r_{D}=\rho_{m}/\rho_{D}$ is the Debye shielding parameter. It is the ratio of the mean distance between ions $\rho_{m}={\left(\frac{4}{3}\pi n_{e}\right)}^{-1/3}$ and the Debye radius $\rho_{D}=\sqrt{\frac{\varepsilon_{0}k_{B}T_{e}}{e^{2}n_{e}}}$. $r_{D}\approx 8.899\times 10^{-3}n_{e}^{1/6}T_{e}^{-1/2}$. *κ* is a constant equal to 1 for a neutral emitter and 1.6 for an ion emitter. Equation (14) reads:(16)$$\Delta \lambda_{s}=2\times 10^{-22}\left[1+5.534\times 10^{-6}n_{e}^{1/4}a\left(T_{e}\right)\left(1-6.742\times 10^{-3}\kappa n_{e}^{1/6}T_{e}^{-1/2}\right)\right]\times n_{e}w_{e}\left(T_{e}\right).$$

### 3.3. Line Intensity

The intensity $I_{ul}$ of each line is related with the energy of the upper emitting level through the following equation:(17)$$ln\left(\frac{I_{ul}}{A_{ul}h\nu_{ul}}\right)=\left(-\frac{1}{k_{B}T}\right)E_{u}-ln\left(\frac{4\pi Q\left(T\right)}{N_{0}g_{u}V\Omega R\left(\nu \right)}\right)$$

Q(T) is the partition function, which depends on the temperature, and *Ω* is the solid angle through which the emitting volume *V* is observed. R(ν) is the optical response of the device, which is dependent on the wavelength. Therefore, the intensity ratio of two lines is given by:(18)$$\frac{I_{1}}{I_{2}}=\frac{A_{1}g_{u1}R\left(\nu_{1}\right)\nu_{1}}{A_{2}g_{u2}R\left(\nu_{2}\right)\nu_{2}}exp\left(\frac{E_{2}-E_{1}}{k_{B}T}\right)$$

It is useful to introduce the following parameter:(19)$$\rho =\frac{I_{1}^{theo}/I_{ref}^{theo}}{I_{1}^{exp}/I_{ref}^{exp}}$$

By choosing a given reference line, it is possible to determine with Equation (18) the theoretical ratio of the intensity of any other line to the intensity of this one. Its comparison with the same experimental ratio through the parameter *ρ* indicates whether the upper levels are populated according to Boltzmann distribution or not. *ρ* is then expected to be equal to 1. If *ρ* is large, the intensity of the selected line is weaker than expected with regards to the intensity of the reference line.

### 3.4. Applied Method

In this work, the broadening parameters of 12 among 14 selected lines of bismuth were determined (two were too weak to determine their broadening parameters). High-pressure lamps are often preferred in that purpose, as more accurate parameters for line transitions can be obtained. Indeed, the reproducibility of discharges in liquids, and therefore the corresponding dynamics, is usually weak, as discussed hereafter. Here, our main concern was first and foremost the evolution of the plasma parameters in a dense media, where this kind of information is rarely straightforwardly available.

To have access to the best information available, the following strategy was adopted. For a given transition, 21 spectra (each being the average of 50 spectra recorded with an exposure time of 50 ns) were recorded every 50 ns from 50 to 1050 ns. Six spectra were selected among these 21—first, to describe, at best, the time evolution observed on average and, second, to limit the number of data to process. This selection is made on the intensity of the recorded transition; the more pronounced it is, the better it is. It does not take into account the intensity of the background.

The spectra at the shortest times (from 50 to 150 ns) adopt the shape of a continuum emission without visible atomic transition. Due to this, the selected lines could only be fitted from 200 ns to 1050 ns. On this basis, the description of the time evolution of each selected line was subjected to a simple rule: decrease as much as possible the number of degrees of freedom to avoid overparameterization of the problem and get reliable solutions. Time-resolved spectra were thus exploited to check the reliability of the extracted data. The free parameters were then:the basic parameters required in Equation (16)—*w_e_*, *d_e_* and *a*(*T_e_*)—to account for Stark broadening (see Table 2). These three parameters are usually set with one spectrum, as the broadening, the position of the maximum and the shape of a given line fully determines the *w_e_*, *d_e_* and *a*(*T_e_*);the electron density. This variable must decrease in time according to the same law (basically, an exponential decay for a first-order process), whatever the lines;the intensity of the continuum emission. This variable must also decrease in time (basically as $t^{-4/3}$ for a solid body cooling down) and follow the time evolution of the continuum.

## 4. Results and Discussion

The discharge pressure has no significant influence on line broadening, as explained already, and it can be arbitrarily set between 5 and 10 bars without affecting the results. We chose 5 bars, a realistic value at this stage of the discharge evolution [34], i.e., in the time range investigated in this work. Indeed, the pressure decayed quickly. Therefore, initial pressures, ranging from tens to hundreds of bars, dropped in tens of nanoseconds to a few bars only. This time range spanned from 100 to 1050 ns, but it practically reduced from 200 ns to 1050 ns, as lines could not be modelled at short times where the medium was optically thick. The electron temperature was also assumed to be constant, as a change by a factor of ten of the electron density only affects the electron temperature by 10%, according to Saha’s equation. We chose *T_e_* = 0.35 eV. This value came up naturally, higher values leading to unrealistic broadenings. It is lower than electron temperatures usually estimated in the same kinds of discharges that lie rather between 0.5 and 1 eV. The reason for this is due to the conditions that prevail in these kinds of discharges: assuming that the medium is close to the local thermodynamic equilibrium (LTE), Saha’s equation applied to bismuth indicates that if the electron density is 1 × 10^16^ cm^−3^, the electron temperature should be 0.37 eV only, which is compliant with the present estimate. Resorting to Saha’s equation is not illegitimate, even though the discharges in the present conditions were probably closer to equilibrium at shorter times. Discharges in liquids, behaving as arc or spark discharges at high currents, are known to be close to LTE [35,36]. In this case, the emissive states must follow a Boltzmann distribution (depicted in Appendix A for the present case), which is also an assumption needed in the optical model.

By following 14 transitions, we could determine the time evolution of the discharge parameters, as well as the Stark parameters—*w_e_*, *d_e_* and *a*(*T_e_*) (Table 1). The accuracy of the Stark data derived from the present study was first defined by the coherency of the set of time-resolved spectra. This aspect was evaluated qualitatively. The accuracies given in Table 1 ranged from C to E on a scale from A to E as that is used by NIST for line transitions. It is also dependent on the accuracy of the model used to describe the discharge, as it only takes into account a very simplified approach of this complex medium. As no model of discharge in liquids in our conditions has ever been made, quantifying the accuracy of the present values could only be made by comparisons with other published data. However, Stark data for transitions belonging to the Bi I system are not available, to the best of the authors’ knowledge. Only data for transitions belonging to the Bi II system are available.

Concerning the different modelled transitions, two examples were selected here (Figure 3 and Figure 4). Other transitions are available as Appendix A. The following comments can be made:The agreement between experimental values and the theoretical profiles is very satisfactory. It could be even better by changing the parameters that were set once and for all in the treatment process according to the employed method. Line position, asymmetry, reversal and broadening are well-rendered. Of course, the less satisfying profiles are those where the information is the noisiest, i.e., at short times. It is very clear for the transition at 302.46 nm, where the data before 750 ns could not be modelled (Appendix A). On the contrary, the intense isolated lines at 289.80, 351.09, 359.61, 412.17 and 472.25 (Appendix A) were very well-fitted.Minor contributions to a main transition by weak lines (at 307.67 nm for the transition at 306.77 nm (Figure 3), at 340.28 nm and 340.53 nm for the transition at 339.72 nm (Figure 4) and at 299.33 nm for the transition at 298.90 nm (Appendix A) are needed to describe accurately the whole profile. It is also true for the blue wing of the transition at 293.83 to which the red wing of the transition at 289.80 contributes (Appendix A), and the red wing of the transition at 302.46 nm to which the blue wing of the transition at 306.67 nm contributes as well (Appendix A).The fact that the transition at 306.77 nm is resonant (Figure 3) makes a double reversal of the line (i.e., a small bump in the hole of the line), which is not observed for the other transitions. Experimentally, this double reversal is not clear but not unlikely anyway (Appendix A). This double reversal comes from the fact that the photons of the continuum are trapped by a lower level implied in a transition where a hole already exists at its central wavelength because the transition is resonant.All asymmetric lines have their blue wing expanded, except the one at 351.09 nm whose red wing is expanded—Appendix A). The singular behaviour of the transition at 351.09 nm is unexplained.The accuracy of the profiles of the weak lines at 307.67 nm, 340.28 nm and 340.53 nm is naturally limited, even though their presence improves the profile of the set they form with the intense line they neighbour. It is important here to mention that these lines should spring up even more in the spectrum than they do (e.g., Figure 4) if they were not broadened by the Stark effect.

From the previous results, the time evolution of the electron density was deduced (Figure 5). The density decreased from 6.5 ± 1.5 × 10^16^ cm^−3^ at 200 ns to 1.0 ± 0.5 × 10^16^ cm^−3^ at 1050 ns. The spread of the data was limited, which gave a certain confidence in these results.

The intensity of the continuum emission, as a parameter of the model, may be compared with the experimental data. Indeed, this data is needed in the optical model through the term $\varepsilon_{bg}\left(r,\nu \right)$ in Equation (4). Then, we can compare the time evolutions of this normalised parameter and the normalised experimental value. This is an assessment of the reliability of the optical model. For one line at a given time, the background intensity is evaluated as the value of the linear baseline of the peak at the centre of the selected peak. A set of time values for the background intensities of a given line are then derived by this method. This set of time values is normalised and compared next to those used in the model, which are normalised as well. Figure 6 shows the time evolution of the normalised intensity of the continuum for all the selected Bi I transitions at the six chosen times. The normalised experimental values are compared to the parametric values used in the model. Except for the lines at 306.77 nm and 359.61 nm, the agreement is, on average, very satisfactory.

The time evolution of the different lines belonging to the bismuth I and II systems was studied with the 100-gr. mm^−1^ grating between 50 and 550 ns to take into account all the observed lines and benefit from higher intensities. For the sake of clarity, modelled transitions belonging to the Bi I system (from 289 to 472 nm) were presented before the five weak UV (from 262 to 281 nm) and ionic transitions. Close lines (at 298.90 and 299.33 nm; at 306.77 and 307.67 nm and at 339.72, 340.28 and 340.53 nm) were treated here as single lines, only the parameters of the most intense line of the set being considered.

The results concerning the transitions from 289 to 472 nm in the Bi I system are shown in Figure 7. In Figure 7A, the time evolution of the intensity of each line is depicted. Except the transitions at 412.17 and 472.5 nm that reach their maximum after 400 ns, all the other lines behaved similarly, with a maximum at 300 ns.

In Figure 7B, the time evolution of the ratio of the intensity of a given line X to the intensity of the line at 339.72 nm is given. It is used to compare the time evolution of the most energetic transitions (at low wavelengths) to the less energetic transitions (at high wavelengths). It is quite constant and relatively close to 1 for all transitions in time, confirming the negligible variations of the electron temperature. The evolution at 200–250 ns of the group of five lines within the wavelength range (289–307 nm), where *I*(X)/*I*(339.72) exceeded 3, was due to a higher continuum emission than expected. This could be caused by a slightly higher electron temperature during these specific measurements. We also noticed that all data at 350 ns were weaker than expected, probably also for a reason of weak reproducibility intrinsic to this process.

A comparison with the intensity ratios determined by assuming a Boltzmann distribution from the available data led to an acceptable agreement (the ratio *ρ* between the theory and experiment was between 0.25 and 1.51) for all lines but two: the one at 306.77 nm (*ρ* = 158) and the one at 472.14 nm (*ρ* = 50.1). The most striking feature about these two lines was that they were emitted from the resonant level lying at 4.04 eV. Experimentally, the measured intensity was then much lower than expected. The apparent lifetime of the state was increased by the optical thickness of the medium, which could explain the lowering of the intensity of the transitions coming from the 6p^2^ (^3^P_0_) 7s_1/2_ level.

The five UV lines from 262.79 nm to 280.96 nm (Appendix A) were not modelled, as their intensities were too weak with the 1800-gr. mm^−1^ grating. The time evolution of the intensity of these lines, established with the 100-gr. mm^−1^ grating, showed that the maximum was reached at about 300 ns (Appendix A), like for most of the other neutral lines.

The comparison with the intensity ratios determined by assuming a Boltzmann distribution from available data (inset in Appendix A; a reference line was taken at 262.79 nm) led to an acceptable agreement (the ratio *ρ* between the theory and experiment was between 1.2 and 2.6) for all the lines but the one at 273.05 nm (*ρ* = 33.13). For this latter, the adopted *A_ul_* value was 2.02 × 10^7^ s^−1^ (Table 1). This value was determined by Stanek et al. [37]. To get the same agreement as the other lines, it should be about 2 to 3 × 10^8^ s^−1^. As the upper level of this transition was very close to the ionization potential (7.225 eV versus 7.285 eV), the lower intensity observed experimentally could possibly be due to an easy loss of this state by ionisation.

The two weak ionic lines at 379.25 nm and 430.17 nm were treated alike (Appendix A). The time evolution of the line at 430.17 nm showed that the maximum intensity was reached after 450 ns and at 250 ns for the other ionic line, demonstrating different excitation mechanisms. The line intensities were nevertheless proportional until 250 ns (Appendix A). This observation cannot be exploited by the lack of data for *A_ul_* in these transitions.

## 5. Conclusions

The derivation of Stark data for Bi I lines is a convenient way to access discharge parameters, even though the obtained data is not as accurate as it would be with more conventional methods. The resulting values of the parameters show that the discharge is close to the thermodynamic equilibrium.

The possibility of using an optical fibre directly in the liquid opens up the way to new perspectives. The recording of enhanced intensity signals improves the quality of experimental results and may bring complementary new information. In this first attempt, the presence of lines belonging to the Bi II system was evidenced, which could not be observed otherwise. Further experiments could benefit from this possibility to access spatially resolved emissions.

## Figures and Tables

**Figure 1 molecules-26-07403-f001:**
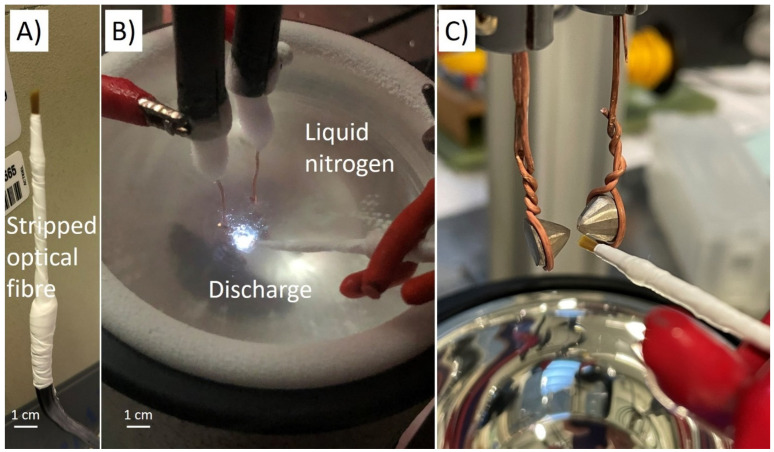
(**A**) Picture of a stripped multi-strand optical fibre. (**B**) Picture of the discharge in liquid nitrogen. (**C**) Picture of the bismuth electrode (the gap distance is set at 100 µm typically during the process—the gap is larger for the picture).

**Figure 2 molecules-26-07403-f002:**
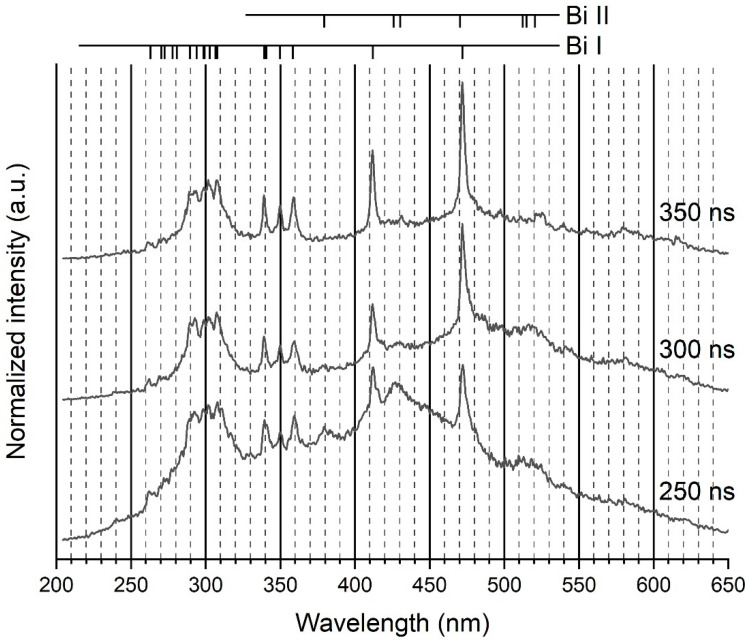
Example of three recorded spectra over a range of wavelengths (200–650 nm). Ionic lines are clearly visible, despite their weakness. The grating was 100 gr. mm^−1^. Integration time: 50 ns.

**Figure 3 molecules-26-07403-f003:**
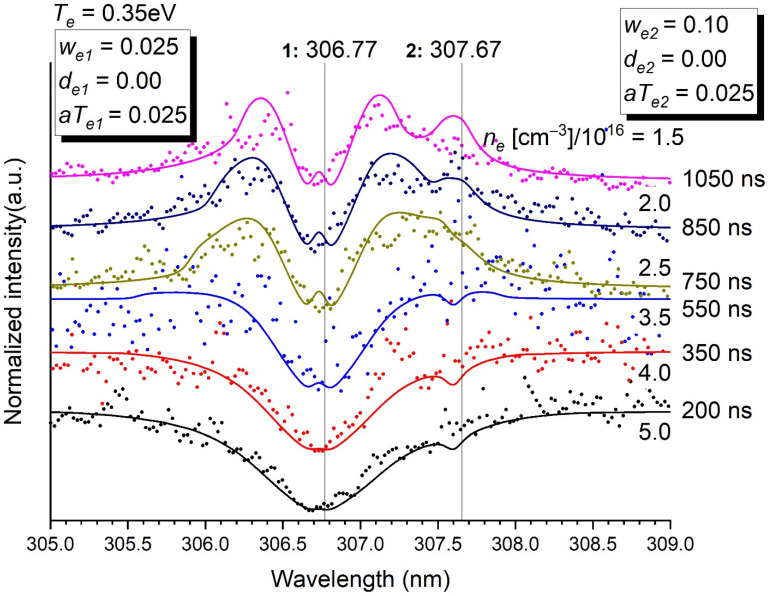
Normalised intensity of the emission of the transitions at 306.77 nm (resonant) and 307.67 nm as a function of time. Stark parameters are given in the inset. The electron density *n_e_* in (cm^−3^)/10^16^ is determined for each time. *T_e_* = 0.35 eV. *p* = 5 bars.

**Figure 4 molecules-26-07403-f004:**
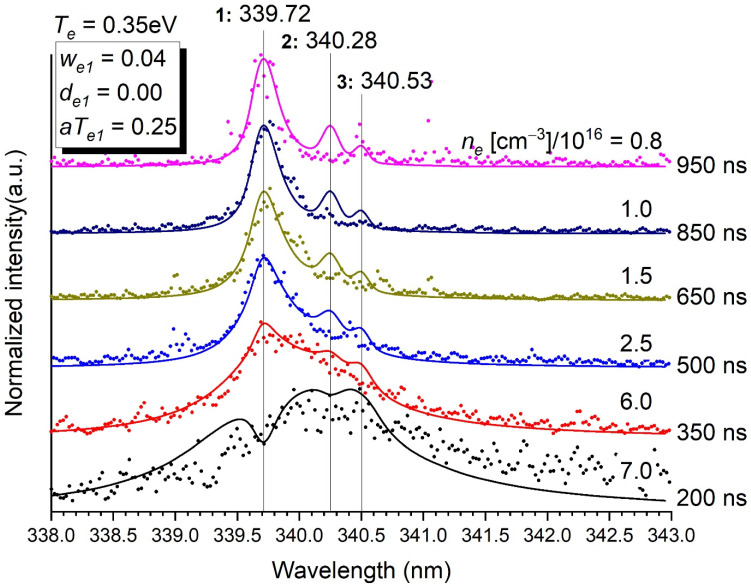
Normalised intensity of the emission of the transitions at 339.72 nm, 340.28 nm and 340.53 nm as a function of time. Stark parameters are given in the inset. The data at 250, 300 and 550 ns are too noisy to be exploited. The electron density *n_e_* in (cm^−3^)/10^16^ is determined for each time. *T_e_* = 0.35 eV. *p* = 5 bars. Stark parameters for lines 2 and 3 could not be determined with sufficient enough accuracy to be reported here.

**Figure 5 molecules-26-07403-f005:**
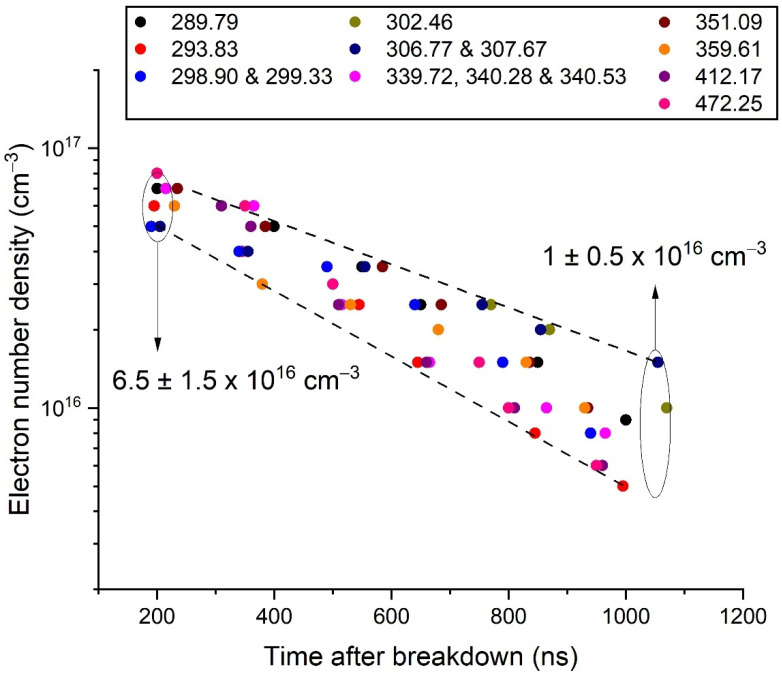
Evolution of the electron number density versus time after breakdown, as determined by the modelling of some selected Bi I transitions. The dots are slightly shifted in time (from −20 ns to +20 ns) to make them visible.

**Figure 6 molecules-26-07403-f006:**
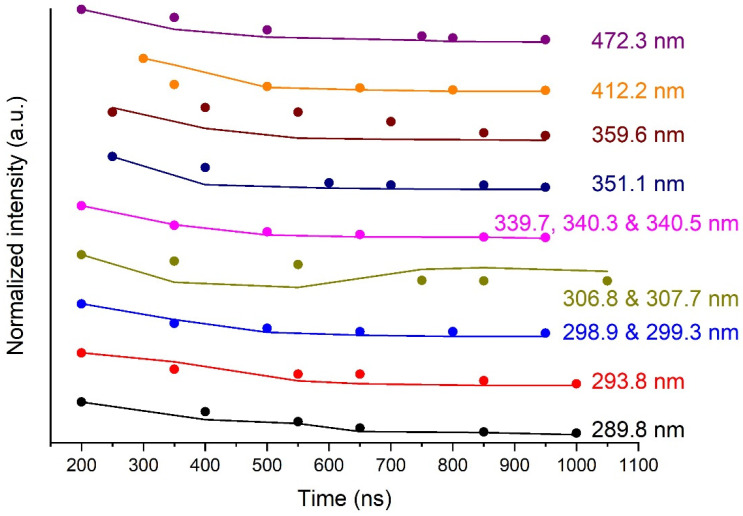
Dots: normalised parametric intensity of the continuum as a function of time for all the modelled Bi I transitions at the 6 chosen times. Strait lines: idem for the normalised experimental intensity. 1800-gr. mm^−1^ grating. Integration time: 50 ns.

**Figure 7 molecules-26-07403-f007:**
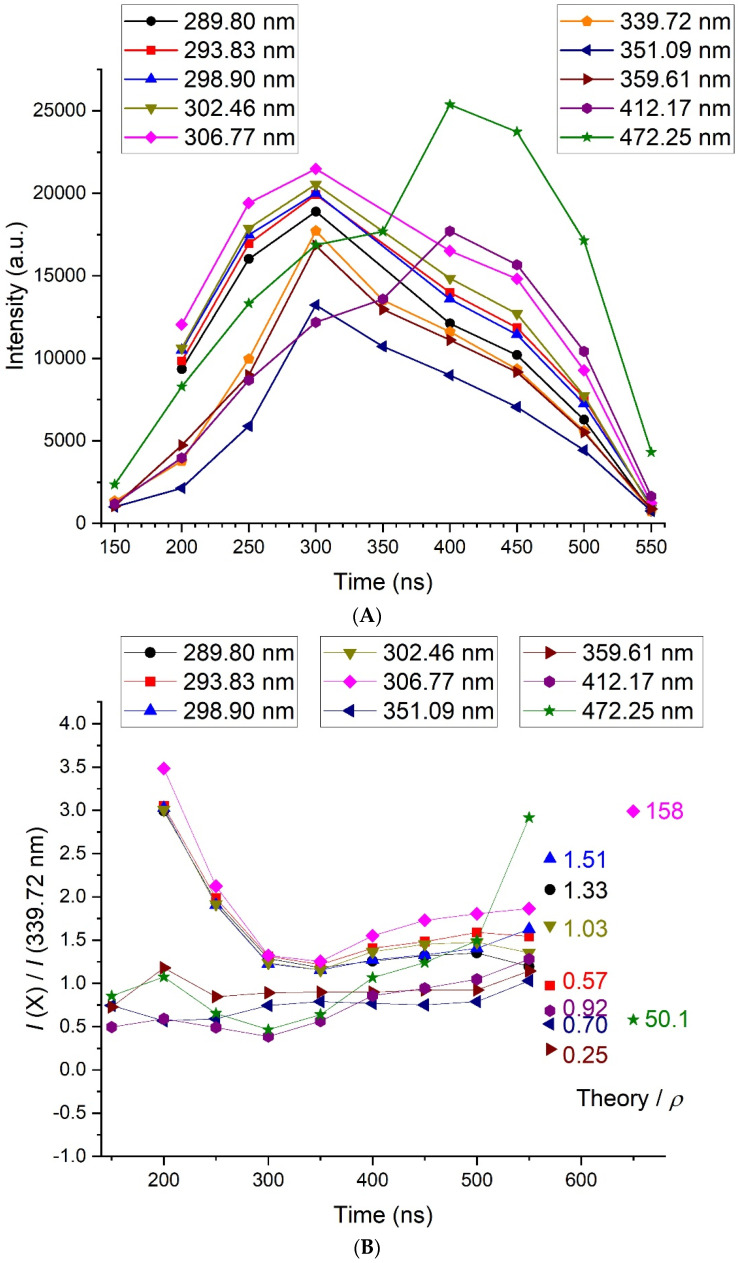
(**A**) Time evolution of the intensity of the selected Bi I neutral lines, as measured with the 100-gr. mm^−1^ grating. (**B**) Time evolution of the ratio of the intensity of a given line X to the intensity of the line at 262.79 nm. The dots under “Theory” (put arbitrarily at 570 ns and 650 ns for outliers) correspond to the average expected value for a Boltzmann distribution and can be read on the y-scale. *ρ* (given by numbers between 0.25 and 158) is the ratio between the theoretical ratio and the average experimental ratio.

**Table 1 molecules-26-07403-t001:** Basic data used in the model and taken from References [26,27]. Acc. is the accuracy of the line (same meaning as NIST). Aul is the transition probability. *E_low_* and *E_up_* are the energies of the lower and upper levels. *J_low_* and *J_up_* are the total electronic angular-momentum quantum numbers of the lower and upper levels. *C’_3_* values are deduced from the oscillator strengths taken in Reference [24] and Equation (8), and *C’*_6_ values are calculated using Equation (10). *w_e_*, *d_e_* and *a*(*T_e_*) are the data needed in Equation (16) (see text for details).

	Wavelength	Rel. Int.	Obs.	Acc.	*A_ul_*	*E_low_*	*E_up_*	*J_low_*	*J_up_*	*C’* _3_	*C’*_6_ *	*w_e_*	*d_e_*	*a*(*T_e_*)
	(nm)	(/100)	Int.		(s^−1^)	(eV)	(eV)			(m^3^/s)	(m^6^/s)	(nm)	(nm)	
Bi I system	262.7904	160	w		4.56 × 10^7^	1.4157804	6.1323713	1	1		1.12 × 10^−41^			
	269.6748	50	vw		6.20 × 10^6^	1.4157804	6.0119775	1	2		9.10 × 10^−42^			
	273.0492	17	vw		2.02 × 10^7^	2.6856111	7.2249911	0	1		1.63 × 10^−39^			
	278.0476	50	vw		3.09 × 10^7^	1.4157804	5.8735626	1	0		7.32 × 10^−42^			
	280.9615	25	vw		1.34 × 10^7^	1.9140062	6.3255610	2	1		1.63 × 10^−41^			
	289.7964	410	s	D+	1.53 × 10^8^	1.4157804	5.6928439	1	0		5.66 × 10^−42^	0.01	0.00	0.35
	293.8297	280	s	D+	1.20 × 10^8^	1.9140062	6.1323713	2	1		1.11 × 10^−41^	0.05	0.00	0.35
	298.9004	280	s	D+	5.40 × 10^7^	1.4157804	5.5625605	1	1		4.69 × 10^−42^	0.01	0.00	0.025
	299.3327	60	s	D+	1.45 × 10^7^	1.4157804	5.5565801	1	2		4.74 × 10^−42^	0.05	0.00	0.025
	302.4618	340	s	D+	8.62×10^7^	1.9140062	6.0119775	2	2		9.02 × 10^−42^	0.025	0.00	0.40
	306.7699 **	2300	s	D+	1.67 × 10^8^	0	4.0404245	1	0	9.72 × 10^−15^	1.18 × 10^−42^	0.025	0.00	0.025
	307.6654	30	w	D	3.31 × 10^6^	1.4157804	5.4469162 ^†^	1	1 ^‡^		1.02 × 10^−42^	0.1	0.00	0.025
	339.7198	70	s	D+	1.79 × 10^7^	1.9140062	5.5625605	2	1		4.69 × 10^−42^	0.04	0.00	0.25
	340.2800	9	vw	E	1.50 × 10^6^	1.9140062	5.5565801	2	2		4.64 × 10^−42^	***	***	***
	340.5330	17	vw	E	6.40 × 10^6^	2.6856111	6.3255610	0	1		1.61 × 10^−41^	***	***	***
	351.0864	60	s	D+	6.50 × 10^6^	1.9140062	5.4444405	2	1		4.04 × 10^−42^	0.04	0.00	0.25
	359.6110	27	s	D+	1.93 × 10^7^	2.6856111	6.1323713	0	1		1.09 × 10^−41^	0.05	0.00	0.20
	412.1704	26	s	D+	1.64 × 10^7^	2.6856111	5.6928439	0	0		5.38 × 10^−42^	0.05	0.10	0.45
	472.2528	110	vs	C	9.40 × 10^6^	1.4157804	4.0404245	1	0		1.02 × 10^−42^	0.025	0.04	0.45
Bi II system	379.2564	70	w			9.8060517	13.0742633	2	3					
	425.9413	75	vw			10.1985993	13.1086088	3	4					
	430.1697	70	w			10.1728579	13.0542630	2	3					
	470.5285	60	hd			10.4494435	13.0837049	1	2					
	512.4356	50	vw			11.0062562	13.425090	2	3					
	514.4492	60	vw			8.5715101	10.9808694	0	1					
	520.9325	75	vw			8.6291111	11.0084923	1	2					

* Collider: N_2_, ** resonant line, vw: very weak, w: weak, s: strong, vs: very strong, hd: hidden, *** too weak to be estimated, ^†^ unknown level and ^‡^ assumed. The estimated accuracies for the Stark parameters are: C < 25%, D+ < 40%, D < 50% and E > 50%.

**Table 2 molecules-26-07403-t002:** Broadening contributions expressed in pm for different Bi I transitions due to different sources. *T_e_* = 0.35 eV. *p* = 5 bars. *n_e_* = 6 × 10^16^ cm^−3^. vdW IA: van der Waals impact approximation. QS: quasistatic approximation.

Wavelength	Instrument	Doppler	vdW IA	vdW QS	Resonance	Stark QS
289.7964	100	0.91	3.8	0.5	0	159
293.8297	100	0.92	5.2	1.1	0	794
298.9004	100	0.94	3.8	0.5	0	123
299.3327	100	0.94	3.8	0.5	0	614
302.4618	100	0.95	5.0	0.9		404
306.7699	100	0.96	2.3	0.1	93.6	307
307.6654	100	0.97	2.2	0.1	0	1228
339.7198	100	1.07	4.9	0.6	0	591
340.2800	100	1.07	4.9	0.6	0	368
340.5330	100	1.07	4.9	0.6	0	123
351.0864	100	1.10	8.6	2.3	0	591
359.6110	100	1.13	7.7	1.6	0	711
412.1704	100	1.30	7.6	1.0	0	850
472.2528	100	1.49	5.1	0.2	0	425

## Data Availability

Data can be provided on demand to the corresponding author.

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
