# Peer review of "Study by Optical Spectroscopy of Bismuth Emission in a Nanosecond-Pulsed Discharge Created in Liquid Nitrogen"

_molecules, 2021, doi:10.3390/molecules26237403_

Round 1

Reviewer 1 Report

Dear authors

Many thanks for your contribution. Results seems be interesting and valuable for the field. I have some comments (see below) that should be considered also for your future works in the field. I have also some note because I’m not fully sure if selected journal is the best fitted for the topic. But if we decide that atom is a special kind of molecule, this is OK. Here is a list of my comments.

  1. Fig. 1C – may be, it will be better to show electrodes with fiber without liquid nitrogen. Such picture will be better readable.
  2. Was observed production of something (NPs or coating on electrodes holder or optical fiber) after longer discharge operation? Deposition of bismuth layer on fiber can affect spectra.
  3. Use of grating blazed at 330 nm up to 900 nm seems be non-realistic because such grating have commonly very low reflectivity over 700 nm. Also detection of lines bellow 300 nm is disputable with 100 lines grating due to the same effect.
  4. I’m not sure if all line parameters presented in Table 1 are necessary for the paper understanding. This is generally copy from tables, so I’m recommending getting simply information about lines position (or as identified spectrum picture) with link to the supplementary material.
  5. The expected line shifts seem be close to the spectrometer accuracy. Probably, you tried this accuracy using some low pressure atomic lines source. Similar shifts can be observed also due to simple temperature changes in laboratory.
  6. Is pressure of 5 bars used in Table 2 realistic? It is known that in pulsed discharges extremely strong shock waves are generated with pressure couple orders higher. But maybe, they are out of region of interest in times when spectra are collected. But maybe, these waves are reflected by the discharge vessel walls and thus they are existing longer time.
  7. May be, it will be possible to try compare electron temperature and electron density used in this study with date collected if some well-known lines getting gas (like helium or hydrogen) will be dissolved in liquid nitrogen.
  8. Figure 4 – there is pointed out in the caption that data at 350 ns are too noisy but in picture they seem be without significantly higher noise then others. May be, you was thinking about another time.
  9. Use also different symbols for different lines for their better distinguishing. Color differences can be not so well visible. Why right legend in Fig 7B is rather chaotic? Similar comment is also for Fig. 6 in the supplementary material.

Author Response

Please, see attached word file.

Regards,

Th. Belmonte

Reviewer 2 Report

The paper presents a spectroscopic study of nanosecond pulsed plasma in liquid nitrogen between Bi electrodes. Such plasmas are very complex objects relevant to numerous applications, therefore I believe that this original work should be accepted after the following points are addressed:

  1. Only Bi I and Bi II emission is considered in the analysis while N2 molecular emission is neglected. The possible contribution of the N2 emission should be discussed.
  2. Page 2 line Is the initial sharpness of 1 micrometer pins preserved during the experiment? A micrograph of the pin-electrodes should be shown before and after the experiment.
  3. Page 2, line 59. Power dissipation per pulse between 1 and 5 mW looks strange to me. Do the Authors mean energy per pulse measured in mJ? Such significant pulse-to-pulse scatter of the input energy and its impact on the experimental results should be discussed in more detail.
  4. Table 1 header – what is the meaning of spectroscopic parameters in the last five columns? This should be more clearly explained in the text.
  5. Page 5 line 151. The assumption of Gaussian radial profiles of electron and excited species densities with identical FWHM should be justified. Such profiles are not physical because they do not  take into account radial gradients of electron temperature and gas pressure. Moreover, such radial profile is not applicable to the Bi I ground state.
  6. Page 8 line 270. The assumption of constant Te=0.35 eV should be justified. If LTE conditions are satisfied, Te should vary in time in accordnace with the variation of ne. It is claimed in the Abstract that the electron temperature is found to be 0.35 eV, I would say that the Te is arbitrarily set without any justification.
  7. Figure 4 emission lines at 340.28 nm and 340.53 nm don’t seem to be present in experimental spectra. The fit for t>500 ns is not good and Stark parameters for these lines should not be reported unless the fit is improved.
  8. The overall clarity of the text from line 329 till line 420 should be improved. I did not understand the purpose of figures 6-7. What conclusions can be drawn from the data? More specifically:
  9. Fig 6A how was the intensity normalized? What was the procedure to determine experimental continuum intensity, how was the spectral line intensity subtracted? How was the parametric intensity obtained? What do we learn from this figure?
  10. Fig 6b and the corresponding text (lines 335-339). Not clear, should be better explained.
  11. Fig 7b and the corresponding text (lines 357-364). Not clear, should be better explained.
  12. A more detailed analysis of the derived Stark data is needed. What is the accuracy of reported parameters, how do they compare with the available literature data for Bi?

Round 2

Reviewer 1 Report

  1. 1C – may be, it will be better to show electrodes with fiber without liquid nitrogen. Such picture will be better readable.

Our Answer: We made the modification.

Many thanks, now picture is clear.

  1. Was observed production of something (NPs or coating on electrodes holder or optical fiber) after longer discharge operation? Deposition of bismuth layer on fiber can affect spectra.

Our Answer: Indeed ! Nanoparticles of Bi are produced by this process. We can collect them by sedimentation on the bottom of the vessel. However, there is no deposit on the fiber. This is certainly due to the emission of shock waves that clean up the interelectrode gap and its vicinity by pushing the NPs away. We introduced a short explanation to clarify this point: "Though nanoparticles are produced by electrode erosion, deposition on the fibre is negligible. It is certainly because of shockwaves emitted during the very early stage of the discharge process that push nanoparticles away from the interelectrode gap."

I’m thinking that NPs should have some interesting properties and may be, they will be applicable for some additional processes like further synthesis of alloys or shell structures. Also it will be interesting if nitrogen is bult up in the particles. May be not because you didn’t detect nitrogen lines.

  1. Use of grating blazed at 330 nm up to 900 nm seems be non-realistic because such grating have commonly very low reflectivity over 700 nm. Also detection of lines bellow 300 nm is disputable with 100 lines grating due to the same effect.

Our Answer: When a grating is blazed at 500 nm for instance, it only means that the maximum of the optical response stands at this very wavelength. But the response of the grating at other wavelengths can be sufficient and if transition intensities are strong, they can be detected (see graph below). And they are as spectra show it.

You’re partially right. Officially by manufactures (Jobin Yvon) is declared that applicable reflectivity of grating is between 2/3 up to 2 of blazed wavelength. But as it is my experience, this should be something more. Of course, this also depends on individual grating and its manufacturing technique. It will be probably better to use another grating blazed at lower wavelength for future experiments because you did not detect radiation above 500 nm.

  1. I’m not sure if all line parameters presented in Table 1 are necessary for the paper understanding. This is generally copy from tables, so I’m recommending getting simply information about lines position (or as identified spectrum picture) with link to the supplementary material.

Our Answer: The second referee asks for a clarification of the content of the last 5 columns that are results from the present study and not simply a copy from other tables. We think it would be better to keep it, to give access to parameter values more easily to readers if they need it for new calculations.

OK, not it is clear for me. May be, you can point out in the text, that some columns are not from literature.

  1. The expected line shifts seem be close to the spectrometer accuracy. Probably, you tried this accuracy using some low pressure atomic lines source. Similar shifts can be observed also due to simple temperature changes in laboratory.

Our Answer: I’m not sure to have clearly understood this remark as the line shift due to room temperature is almost null (lower than fm !). Were you meaning line broadening? In this case, A change of +/- 20°C would hardly affect it too (it would be due to the possible dilatation of blades from the entrance slit, and this is negligible too.)

This accuracy was mentioned mainly due to temperature dilatations of spectrometer because they should be rather high. Other effects are really negligible and fully under the used spectrometer resolution.

  1. Is pressure of 5 bars used in Table 2 realistic? It is known that in pulsed discharges extremely strong shock waves are generated with pressure couple orders higher. But maybe, they are out of region of interest in times when spectra are collected. But maybe, these waves are reflected by the discharge vessel walls and thus they are existing longer time.

Our Anwer: It is realistic. The pressure decays quickly. So, initial pressures, ranging from tens to hundreds of bars, drop in tens of nanoseconds to a few bars. Initial pressures can even be higher when there is no initial bubbles in which the breakdown occurs, like it is commonly observed in laser processes. As the time range selected in this work spans from 100 to 1050 ns, this value is realistic, even though it must regarded as an average value, of course.

I agree. May be, you should add this detail at the beginning of Results for clarification.

  1. May be, it will be possible to try compare electron temperature and electron density used in this study with date collected if some well-known lines getting gas (like helium or hydrogen) will be dissolved in liquid nitrogen.

Our Anwer: We tried to know the solubility of He and H2 in liquid N2. I found the following reference: V. Mank, I. Matyash and M. Starkov (1966). Solubility of hydrogen in liquid nitrogen and of helium in liquid hydrogen as given by data on nuclear magnetic resonance(Nuclear magnetic resonance method for studying solubility of helium and hydrogen in liquid nitrogen). but I could not obtained it in a short period of time. I could find anyway a correlation for helium in liquid nitrogen. The helium mole fraction in liquid nitrogen is given by ? = ? ∗ ∙ ???(−9.727 + 0.0472 ∙ ?(?) − 0.025 ∙ ? ∗ ) in , NASA/TM—2012-217716 1, "Correlation of Helium Solubility in Liquid Nitrogen", Neil T. Van Dresar and Gregory A. Zimmerli, National Aeronautics and Space Administration, Glenn Research Center where p* is given by ? ∗ = (? − ?0 )⁄????. P is the pressurization by helium, P0 is the saturated vapor pressure of pure LN2 at temperature T and Pref = 1 MPa. At 77 K, P0 = 0.1 MPa. (Be careful, the formula is wrongly written in the publication !) If P = 17.2 bars, one finds x = 3.52×10-3 . So, we agree, with x = 3.52×10-3 , it should be possible to dissolve sufficient helium in liquid nitrogen. Nevertheless, because we do not see any nitrogen lines, we wonder how we could see helium lines. Furthermore, 20 eV are necessary to excite the first excited level. So, we doubt this could work with helium in our conditions. If hydrogen solubility is high enough too, it might be more likely. Anyway, we will try, in forthcoming experiments, to check whether it works or not.

OK, I was thinking this about idea for the future experiments. Additionally, hydrogen or helium should be introduced into discharge zone as bubbles. But I’m not sure if the discharge conditions will be similar as now. I’m afraid that not because gas will change conditions too much.

  1. Figure 4 – there is pointed out in the caption that data at 350 ns are too noisy but in picture they seem be without significantly higher noise then others. May be, you was thinking about another time.

Anwer: Indeed, it was a mistake. It should read 300 ns and not 350. This mistake has been corrected.

  1. Use also different symbols for different lines for their better distinguishing. Color differences can be not so well visible. Why right legend in Fig 7B is rather chaotic? Similar comment is also for Fig. 6 in the supplementary material.

Anwer: We modified Fig. 7 to include different symbols for different lines. We did not change other figures as curves are well separated. The caption is "chaotic" because dots are not positioned randomly but at the Y-value they adopt. Then, rho-values are written aside.

This should be also applied in Fig. 5 where are too many dots of different color.

In. Fig. 7b I see still disordered legend. May be it was during some pdf conversion. Check this carefully because now some rho numbers are nearly overlapping and they are in two columns.

Author Response

Thanks for your positive comments.

Reviewer 2 Report

The questions raised by the referee have been adressed by the Authors.
